# CHAIN OF INTERACTION BENCHMARK (COIN): WHEN REASONING MEETS EMBODIED INTERACTION

## ABSTRACT

Generalist embodied agents must perform interactive, causally-dependent reasoning, continually interacting with the environment, acquiring information, and updating plans to solve long-horizon tasks before they could be adopted in real-life scenarios. For instance, retrieving an apple from a cabinet may require opening multiple doors and drawers before the apple becomes visible and reachable—demanding sequential interaction under partial observability. However, existing benchmarks fail to systematically evaluate this essential capability. We introduce **COIN**, a benchmark designed to assess interactive reasoning in realistic robotic manipulation through three key contributions. First, we construct **COIN-50**: 50 interactive tasks in daily scenarios, and create **COIN-Primitive** required by causally-dependent tasks, and **COIN-Composition** with mid-term complexity for skill learning and generalization evaluation. Second, we develop a low-cost mobile AR teleoperation system and collect the COIN-Primitive Dataset with 50 demonstrations per primitive task (1,000 in total). Third, we develop systematic evaluation metrics about execution stability and generalization robustness to evaluate **CodeAsPolicy**, **VLA**, and language-conditioned **H-VLA** approaches. Our comprehensive evaluation reveals critical limitations in current methods: models struggle with interactive reasoning tasks due to significant gaps between visual understanding and motor execution. We provide fine-grained analysis of these limitations.

## 1 INTRODUCTION

Recent advances in large-scale pretraining NVIDIA et al. (2025b); Black et al.; Brohan et al. (b) and the creation of diverse datasets O'Neill et al. (2024); Khazatsky et al. (2024) and benchmarks Zhang et al.; Li et al. (2024); Liu et al. have significantly advanced robotic manipulation capabilities. However, current benchmarks primarily focus on simplified tasks that fail to capture the complexity of real-world manipulation challenges, particularly those requiring interaction and causal reasoning over long time horizons in partially observable environments.

Consider a robot tasked with "open a locked door". This seemingly simple instruction requires a sequence of interdependent actions: locating the keyhole, inserting and turning the key, and then rotating the handle with trials for the right directions. Such tasks demand what we term **interactive reasoning**—the ability to continually interact with the environment, gather information, update beliefs, and adapt plans accordingly. This requires multiple capabilities: perceiving partial environmental states, reasoning about causal dependencies between actions, maintaining memory of previous interactions, and dynamically adjusting strategies based on feedback. This capability remains beyond the reach of most current Vision-Language-Action (VLA) models and VLM-based planning approaches.

To address this gap, we introduce **COIN** (Chain Of INteraction) Benchmark, consisting of three complementary components: **COIN-50**, featuring 50 interactive reasoning tasks grounded in everyday activities (with one demonstration per task); **COIN-Primitive**, containing 20 fundamental manipulation skills that serve as building blocks (with approximately 50 trajectories per task); and **COIN-Composition**, bridging **COIN-Primitive** and **COIN-50** for evaluating the robustness of VLA learning across visual understanding and instruction variations. Unlike previous benchmarks that primarily test perception or simple manipulation, our tasks are systematically organized according to a taxonomy of reasoning capabilities required in partially observable environments. Based on

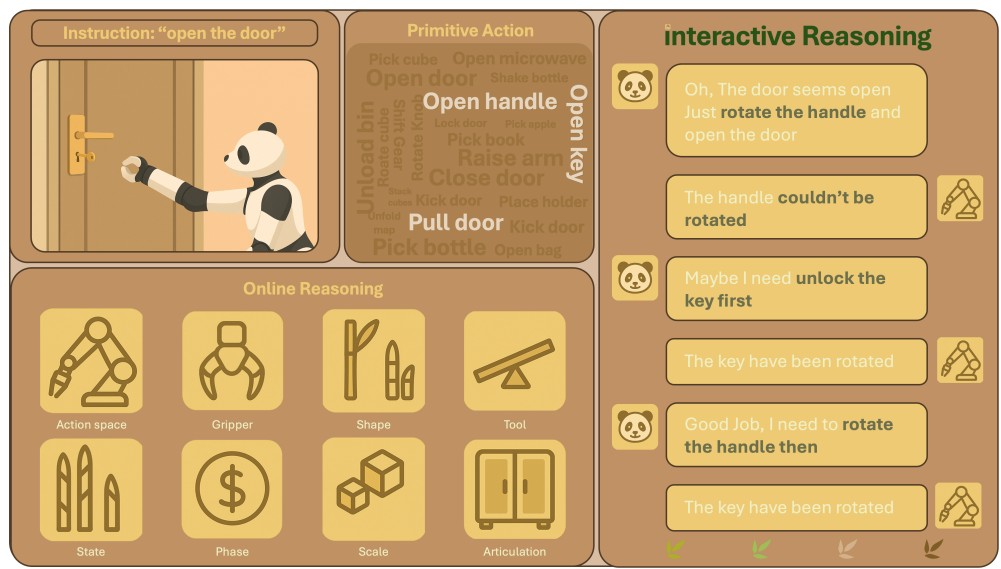

Figure 1: An illustration of **COIN**. Our benchmark focuses on evaluating the crucial **interactive reasoning** ability of Vision-Language-Action (VLA) models and VLM-based robotic planning systems, covering both rich reasoning knowledge and diverse primitive actions.

our analysis, we categorize these capabilities into three principal domains: (1) **Object-Centric Reasoning**, encompassing physical property inference, spatial reasoning, mechanism understanding, and visual reasoning; (2) **Robot-Centric Reasoning**, covering control optimization and embodiment awareness (such as collision handling); and (3) **Compositional Reasoning**, including tool-mediated problem solving, failure-driven adaptation, hierarchical planning, and experience utilization. These capabilities, essential for robots to function effectively in human environments, remain underexplored in existing benchmarks.

To support algorithm development and evaluation, we created a low-cost, phone-based teleoperation system (hardware cost under $20 according to second-hand websites in China) inspired by Rayyan (2024). Using this system, we collected the COIN-Primitive Dataset—over 1,000 expert demonstrations across 20 fundamental manipulation skills recorded from multiple viewpoints. These primitives serve as essential building blocks for VLA model fine-tuning and compositional task solving. **Our contributions include:**

1. **COIN Benchmark**: We construct **COIN-50** with 50 interactive tasks in daily scenarios, **COIN-Primitive** with 20 fundamental manipulation skills required by causally-dependent tasks, and **COIN-Composition** with mid-term complexity for skill learning and generalization evaluation, systematically organized according to a principled taxonomy of interactive reasoning capabilities.

2. **Low-Cost Mobile AR Teleoperation System and Dataset**: We develop a smartphone-based teleoperation system (hardware cost under $20 according to second-hand websites in China) and collect the COIN-Primitive Dataset with 50 demonstrations per primitive task (1,000 in total), enabling accessible data collection for the robotics community.

3. **Systematic Evaluation Metrics and Analysis**: We develop comprehensive evaluation metrics about execution stability and generalization robustness to evaluate **CodeAsPolicy**, **VLA**, and **H-VLA** approaches, revealing critical limitations including significant gaps between visual understanding and motor execution, and provide fine-grained analysis for these limitations.

## 2 RELATED WORKS

### 2.1 VISION-LANGUAGE-ACTION MODELS AND APPROACHES

**CodeAsPolicy** approaches Liang et al. (2022) combine VLMs with predefined skills to orchestrate perception modules Kirillov et al. (2023; 2024); Yang et al. (2023) and low-level controllers in a

| Benchmark | Tasks | Demos | Avg. Steps | Cont. Action | Caus. Dep. | Visual Comp. | Inter. Reas. | Visual Unobs. | Mech. Unobs. |
|---|---|---|---|---|---|---|---|---|---|
| **Robot Manipulation Benchmarks** | | | | | | | | | |
| CALVIN Mees et al. | 34 | 200K | 30 | ✓ | ✗ | ✓ | ✗ | ✗ | ✗ |
| Arnold Gong et al. (2023) | 8 | 40 | 125.8 | ✓ | ✗ | ✗ | ✗ | ✗ | ✗ |
| SimplerEnv Zhu et al. | 10 | N/A | 52.3 | ✓ | ✗ | ✗ | ✗ | ✗ | ✗ |
| Libero Liu et al. | 130 | 50/task | 77.3 | ✓ | ✓ | ✓ | ✗ | ✗ | ✗ |
| VLABench Zhang et al. | 100 | 163 | 157.2 | ✓ | ✗ | ✓ | ✗ | ✗ | ✗ |
| RoboCASA Zheng et al. | 100 | 100/task | 371.9 | ✓ | ✓ | ✓ | ✗ | ✓ | ✓ |
| EmbodiedBench Yang et al. | 100 | N/A | N/A | ✗ | ✗ | ✓ | ✗ | ✓ | ✗ |
| RoboVerse Geng et al. (2025) | 1000 | 9331 | N/A | ✓ | ✓ | ✓ | ✗ | ✓ | ✓ |
| **Vision-Language Reasoning Benchmarks** | | | | | | | | | |
| VLMbench Li et al. (b) | 100 | N/A | N/A | ✗ | ✓ | ✓ | ✗ | ✗ | ✗ |
| ClevrSkills Haresh et al. | 12 | N/A | N/A | ✗ | ✓ | ✓ | ✗ | ✗ | ✗ |
| ReflectVLM Feng et al. | 50 | N/A | N/A | ✗ | ✓ | ✓ | ✓ | ✗ | ✗ |
| **COIN (Ours)** | **90** | **1000+** | **988.9** | ✓ | ✓ | ✓ | ✓ | ✓ | ✓ |

Table 1: Comprehensive benchmark comparison including quantitative metrics and reasoning capabilities. COIN demonstrates the longest average trajectory length (988.9 steps) and uniquely combines all critical reasoning capabilities, particularly interactive reasoning. Our systematic evaluation framework with 1000+ demonstrations across 90 tasks provides unprecedented depth for analyzing interactive manipulation.

modular, zero-shot framework. Works like Huang et al. (a;c;b) excel in generalization but struggle with online adaptation due to their "plan-then-execute" paradigm, where VLMs disengage after initial planning. While recent work Duan et al. introduces replanning mechanisms, significant challenges remain in dynamic, partially observable scenarios requiring continuous interactive reasoning.

**End-to-End VLA models** Brohan et al. (b;a); NVIDIA et al. (2025b); Li et al. (a) directly map visual observations and language to robotic actions via token prediction, learning policies through imitation. Their unified architecture enables emergent reasoning through large-scale pretraining. Despite success in basic manipulation tasks, these models struggle with long-horizon scenarios requiring state maintenance and adaptive planning over extended interactions.

**Hierarchical VLA (H-VLA) architectures** Figure AI (2025); Team et al. (2025) bridge planning and execution by decomposing high-level instructions into subtasks coordinated with low-level executors. This approach combines explicit reasoning with learned behaviors, showing promise in complex manipulation tasks and advancing toward more generalist robotic systems.

## 2.2 ROBOT MANIPULATION BENCHMARKS

**Robot manipulation benchmarks** excel in physical interaction and control capabilities. Works like Arnold Gong et al. (2023) and SimplerEnv Zhu et al. offer photorealistic simulation but lack reasoning components. Libero Liu et al. and RoboCASA Zheng et al. incorporate partial observability, but most lack the combination of dynamic interaction, failure recovery, and interactive reasoning needed for realistic scenarios. Table 1 shows that these benchmarks do not cover interactive reasoning well, while our benchmark emphasizes this crucial capability of embodied AI.

**Vision-language embodied reasoning benchmarks** prioritize reasoning over physical realism. VLMbench Li et al. (b) and ClevrSkills Haresh et al. support causal reasoning in simplified environments, while ReflectVLM Feng et al. offers failure recovery but limited physical interaction. COIN uniquely bridges this gap by combining all eight critical dimensions shown in Table 1, enabling evaluation of true interactive reasoning in realistic, partially observable environments.

## 3 COIN: CHAIN OF INTERACTION BENCHMARK

In this section, we introduce: the formulation of tasks in COIN (3.1), how we built such tasks in COIN (3.2), how we collected datasets with human-in-the-loop teleoperation (3.3), the statistics of COIN (3.4), and the evaluation metrics (3.5).

## 3.1 TASKS FORMULATION

We formalize interactive reasoning tasks as a Partially Observable Markov Decision Process (POMDP): $\mathcal{M} = \langle \mathcal{S}, \mathcal{A}, \mathcal{T}, \mathcal{R}, \mathcal{O}, \mathcal{Z} \rangle$. The state space $\mathcal{S}$ encompasses robot configuration, object states and physical properties. We implement two action spaces, the same as ManiSkill3: for VLA models, $A_{\text{VLA}} = \{\Delta p, \Delta R, g\} \in \mathbb{R}^3 \times SO(3) \times \{0, 1\}$ using delta end-effector poses; for CodeAsPolicy approaches, $A_{\text{VLM}} = \{q_1 ... q_7, g\} \in [q_{\min}, q_{\max}]^7 \times \{0, 1\}$ using absolute joint positions. The transition function $\mathcal{T}$ models state dynamics while the reward function $\mathcal{R} : \mathcal{S} \times \mathcal{A} \to \{0, 1\}$ provides sparse binary success feedback. Observations $\mathcal{O}$ include five camera views (front, left/right back, left front, and wrist-mounted) with depth and segmentation maps, language instructions for the task, and robot proprioceptive data, enabling agents to infer occluded state information through interaction. More details can be found in Appendix C.

## 3.2 TASK BUILDING

**COIN comprises 3 categories and 90 total tasks.** We design a hierarchical task structure that systematically evaluates interactive reasoning capabilities across different complexity levels:

- **COIN-Primitive (20 tasks)**: Fundamental manipulation skills extracted from interactive reasoning tasks by identifying commonly recurring behavioral patterns and essential manipulation primitives (open-close, pick-place, push-pull, rotation).
- **COIN-Composition (20 tasks)**: Mid-term complexity tasks that bridge the gap between primitives and full interactive reasoning, introducing controlled increases in complexity through small visual differences or instruction variations.
- **COIN-50 (50 tasks)**: Full interactive reasoning tasks requiring multi-step causal reasoning under partial observability, where agents must continually interact with the environment to gather information and adapt their strategies.

As shown in Figure 2 and Appendix H.2, we categorized these interactive tasks into three main domains: (1) object information perception and manipulation, (2) robot-understanding and control, and (3) compositional reasoning. This taxonomy helps systematically evaluate different aspects of interactive reasoning capabilities in embodied agents.

**Technical implementation.** We built all environments on the ManiSkill3 platform Tao et al., using the Franka Emika Panda robot in tabletop manipulation settings. Environmental assets include articulated objects from PartNet-Mobility Xiang et al. (2020) and additional assets from Zeng et al.; Li et al. (b); Sketchfab. All 90 tasks provide language instructions and corresponding reward.

**Subtasks and VQA** For each interactive task, we provide: (1) expert demonstrations, (2) ground truth planning in the form of decomposed subtask sequences (*oracle manipulation flow*). As illustrated in Figure 2, the average subtask length is 2.83, highlighting the multi-stage nature of these tasks. (3) VQA (Visual Question-Answering) evaluations. The VQA is similar to ERQA protocol NVIDIA et al. (2025a); Embodied Reasoning (2024) queries VLMs with task-specific questions about success conditions or interaction history, serving as an embodied reasoning probe for future research. We formulated them as multiple-choice problems, where VLMs answer these questions by selecting the right answer.

## 3.3 TELEOPERATION AND DATA COLLECTION

**Low-cost mobile AR teleoperation system.** We introduce COIN-teleoperation, a smartphone-based teleoperation system with a total hardware cost under $20. Built on ARKit Inc. (2023) and ARCore LLC (2023) with the help of Rayyan (2024), this system captures 6-DoF pose data from mobile devices and achieves stable 20Hz control frequency even on older phones (e.g., iPhone 7 Plus), making robotic data collection broadly accessible. Our comprehensive validation G demonstrates 90% data replay success and cross-device compatibility, confirming the reliability of our collection approach.

**COIN-Primitive Dataset.** We collected a comprehensive dataset of 20 **COIN-Primitive** tasks as mentioned in with 50 demonstrations per task captured from 5 camera viewpoints, totaling 1,000 trajectories using COIN-teleoperation. This dataset serves as the primary training resource for VLA

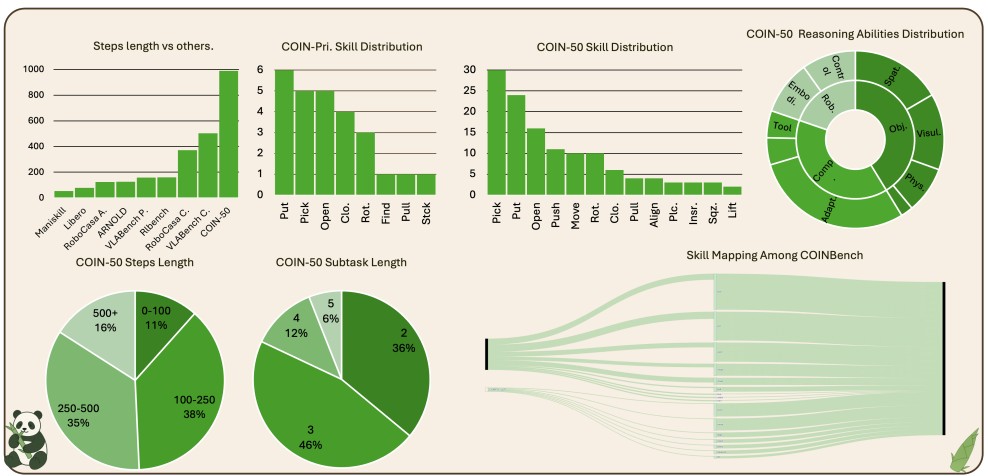

Figure 2: Tasks in COIN: we provide diverse tasks with feasible primitive tasks, and provide GT planning for these tasks, which could be used to guide the planning

model fine-tuning, providing diverse manipulation primitives that form the building blocks for more complex interactive reasoning tasks.

## 3.4 COIN STATISTICS

**Statistics.** Figure 2 presents a comprehensive overview of COIN's benchmark structure. COIN-50 features an average task length of approximately 990 steps, substantially longer than existing benchmarks. More critically, each task requires an average of 2.83 subtasks with frequent interactive reasoning cycles, where 36% of tasks contain 2 subtasks, 46% contain 3, and 12% contain 4. This reveals that our benchmark poses greater challenges not merely through temporal extension, but through the density of reasoning interactions required—necessitating iterative "interaction-reasoning-interaction" loops rather than simple sequential execution, fundamentally distinguishing interactive reasoning from purely long-horizon tasks.

The benchmark's reasoning taxonomy spans object-centric, robot-centric, and compositional reasoning. This focus addresses the under-representation of interactive reasoning in prior benchmarks and supports the modeling of complex "interaction-reasoning-interaction" loops. Overall, COIN offers a comprehensive and realistic testbed for assessing manipulation skills and reasoning capabilities under partial observability and task complexity.

## 3.5 EVALUATION METRICS

We introduce a comprehensive evaluation framework with six complementary metrics that assess different aspects of interactive reasoning and manipulation performance across all COIN tasks:

**Task Performance Metrics:**

- **Success Rate (SR)** measures the proportion of successfully completed trajectories across all evaluated tasks.
- **Class Success Rate (CSR)** measures category-specific performance across reasoning domains (object-centric, robot-centric, compositional).

**Reasoning Assessment Metrics:**

- **Visual Question Answering Score (VS)** assesses perceptual and reasoning capabilities by evaluating whether models correctly answer questions about environmental states and interactive consequences.

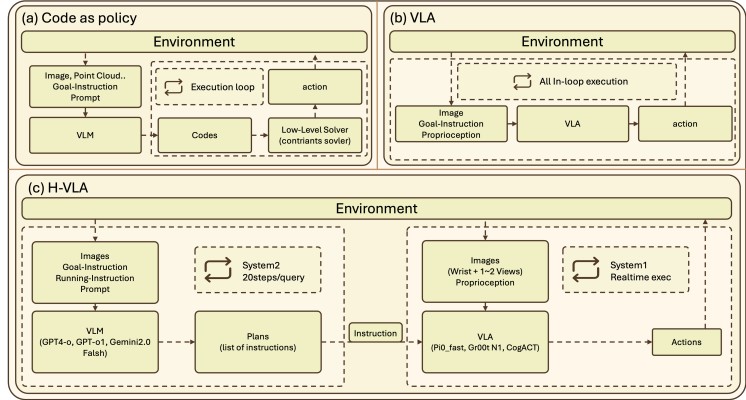

Figure 3: Model Architecture Comparison: **(a)** CodeAsPolicy uses VLMs for planning, with execution handled separately by low-level code and constraint optimizers. **(b)** End-to-End VLA performs in-loop perception and action directly from the environment. **(c)** Hierarchical VLA (H-VLA) combines high-level planning (System 2) with low-level VLA execution (System 1), connected via language instructions.

**Fine-grained Execution Quality Metrics:**

- **Trajectory Stability Score (TS)** measures action quality and smoothness to identify erratic VLA behaviors:

$$TS = 0.3 \cdot S_{vel} + 0.3 \cdot S_{acc} + 0.4 \cdot S_{jerk}$$

  where each component uses $\text{Smooth}(x) = \exp(-CV_x)$ with $CV_x = \frac{\sigma_x}{\mu_x}$ (coefficient of variation) applied to velocity, acceleration, jerk (3rd derivative), and position respectively. Higher scores indicate better trajectory stability.

- **Gripper Control Stability (GS)** assesses manipulation quality through coordination analysis:

$$GS = 0.4 \cdot S_{smooth} + 0.3 \cdot S_{freq} + 0.3 \cdot S_{coord}$$

  where $S_{smooth} = \exp(-\text{abrupt changes})$ penalizes sudden gripper state transitions, $S_{freq} = \exp(-\frac{N_{changes}}{N_{expected}})$ evaluates action frequency appropriateness, and $S_{coord}$ analyzes arm-gripper coordination timing. Higher scores indicate better gripper control quality.

- **Generalization Capability Score (GCS)** evaluates model adaptability through controlled task variations:

$$GCS = \frac{\text{SR}_{\text{composition}}}{\text{SR}_{\text{primitive}}}$$

  where success rates are averaged across all tasks in each category. Scores close to 1.0 indicate good generalization; lower scores reveal generalization failures.

These metrics provide comprehensive evaluation across task completion, reasoning understanding, execution quality, and generalization capability, enabling detailed analysis of model performance across all COIN benchmark components.

### 3.6 HIERARCHICAL VLA (H-VLA) ARCHITECTURE FOR COIN

Similar to Helix Figure AI (2025), we propose a two-layered VLA framework that decomposes complex reasoning tasks into manageable skill sequences, illustrated in Figure 3(c).

- **System 2 (High-level planner)**, a VLM that processes multi-view images and task instructions to generate a sequence of sub-tasks. Operating at fixed intervals, it monitors execution progress by periodically evaluating current observations and adjusting the instruction queue accordingly.

- **System 1 (Low-level executor)**, a VLA model that converts individual skill instructions into robot actions. Taking images, proprioception data, and the current instruction as input, it generates actions in real-time without knowledge of the overall task plan.

## 4 EXPERIMENTS

In this section, we evaluate model performance across different task sets. Section 4.1 introduces our experiment setup and the models tested on COIN (see Figure 3 for an overview). We first analyze how H-VLA and CodeAsPolicy perform on the complex COIN-50 tasks in Section 4.2. Since most models struggle with COIN-50, we then examine their abilities on basic manipulation tasks in COIN-Primitive and COIN-Composition to better understand the causes of failure (Section 4.3).

### 4.1 EXPERIMENTAL SETUP

**Models for COIN-50.** COIN-50's complex interactive reasoning tasks require models capable of adaptive planning and execution. We evaluate:

- **H-VLA models.** As described in Section 3.6, this two-tier architecture combines VLMs for high-level planning with VLAs for execution. We evaluate six configurations pairing two high-level planners (**GPT-4o** and **Gemini 2.0 Flash**) with three VLA models (**Gr00t N1**, **Pi0**, and **CogACT**). Unlike end-to-end VLAs, H-VLA can update plans during execution as new information becomes available through interaction.

- **CodeAsPolicy approaches.** We implement two code-based planning systems: **Voxposer** and **Rekep**, both using `gpt-4o-2024-11-20` for task decomposition and execution planning. Each system reconstructs the environment from three camera views, with Voxposer additionally utilizing ground truth object lists to enhance scene understanding. These approaches separate perception and planning from execution through programmatic interfaces.

**Models for COIN-Primitive and COIN-Composition.** We only consider the "low-level controller" of the two families of models above for COIN-50. Effectively, these are end-to-end VLA models as in H-VLA models (see Section 3.6) and CodeAsPolicy itself, which is the same model on different benchmarks.

- **End-to-end VLA models.** We evaluate 3 cutting-edge vision-language-action models as adopted in H-VLA above: **Gr00t N1** NVIDIA et al. (2025b), **Pi0** Black et al., and **CogACT** Li et al. (a). Both Gr00t N1 and Pi0 process multi-view observations from three cameras (base-front, left-front, and wrist-mounted), while CogACT processes only the left-front view per its design requirements. We fine-tune all VLA models on the COIN-Primitive dataset until convergence or for a maximum of three days (see Appendix F for details). Following Liu et al., we select checkpoints based on validation success rates.

- **CodeAsPolicy approaches.** We evaluate the same Voxposer and Rekep implementations on COIN-Primitive tasks to assess their performance on fundamental manipulation skills.

**Evaluation Details.** We report **SR** averaged over 10 trials. **CSR** is generated from the **SR**, and the VQA score is generated by querying the VLM with expert demonstrations for about 50 steps per query. For **TS** and **GS**, we report the scores according to the recorded trajectories during evaluation. For **GCS**, we evaluate the score according to the **SR** between **COIN-Primitive** and **COIN-Composition**. All tasks and environment specifications can be found in Appendix C and H.2.

### 4.2 MAIN RESULTS FOR COIN-50

**Overview: Interactive reasoning remains a fundamental challenge for current AI approaches.** Our evaluation reveals a stark capability gap in all tested systems when faced with tasks requiring interactive reasoning. As shown in Table 4, all models fail to solve complex interactive reasoning tasks, with success rates rarely exceeding 3%. Our analysis on COIN-50 reveals fundamental limitations in both major approach categories:

**CodeAsPolicy approaches face two critical issues:** (1) *Non-interactive planning architecture*: These methods cannot update plans based on environmental feedback, making them fundamentally unsuited for partially observable environments requiring iterative interaction. For example, in "Pick the cube" task, if the cube was not picked, it only repeat "back-home" and "pick the cube" loop, without any new strategies. (2) *Planning-execution gap*: Significant disconnects exist as shown in

Table 2: Trajectory and gripper stability analysis across different task types. Values show mean $\pm$ standard deviation, revealing execution quality patterns across model architectures. **Bold values** indicate performance exceeding human baseline.

| Model | Task Type | Trajectory Stability | Gripper Stability |
|---|---|---|---|
| CogACT | Primitive | **0.150 ± 0.055** | **0.872 ± 0.134** |
| CogACT | Composition | **0.138 ± 0.039** | **0.796 ± 0.136** |
| CogACT | Interactive | **0.146 ± 0.041** | **0.782 ± 0.141** |
| Gr00t N1 | Primitive | 0.082 ± 0.015 | 0.318 ± 0.116 |
| Gr00t N1 | Composition | 0.086 ± 0.002 | 0.327 ± 0.058 |
| Gr00t N1 | Interactive | 0.084 ± 0.002 | 0.294 ± 0.050 |
| Pi0 | Primitive | 0.084 ± 0.067 | 0.440 ± 0.198 |
| Pi0 | Composition | 0.035 ± 0.043 | 0.465 ± 0.253 |
| Pi0 | Interactive | 0.061 ± 0.050 | 0.440 ± 0.219 |
| Human Dataset | Primitive | 0.134 ± 0.035 | 0.684 ± 0.297 |

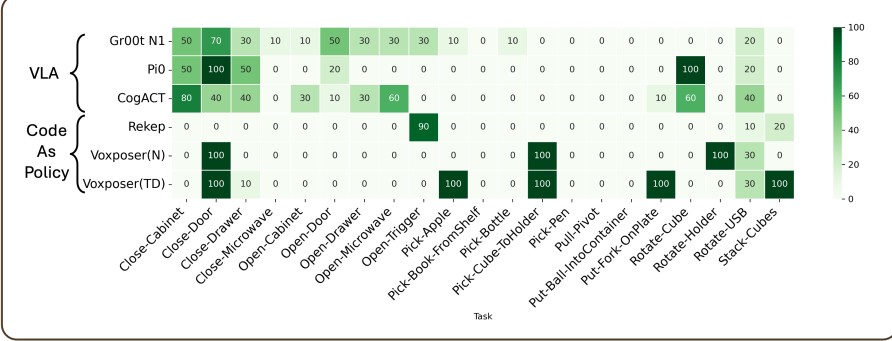

Figure 4: Performance heatmap for different models on COIN-Primitive tasks. The visualization reveals that VLA models achieve broader task coverage than CodeAsPolicy approaches, though with different strengths across task types. Color intensity indicates success rate.

Huang et al. (b) between high-level plans and low-level execution capabilities, which we analyze in detail through COIN-Primitive and COIN-Composition evaluations.

**H-VLA approaches suffer from multiple limitations:** (1) *Poor VLM planning performance*: High-level reasoning and plan generation capabilities are insufficient for complex interactive scenarios, and their performance does not improve significantly as the number of interaction steps increases. (2) *Inadequate VLA execution*: Low-level action generation models demonstrate poor manipulation capabilities. (3) *Weak VLM-VLA integration*: The coordination between high-level planning and low-level execution remains problematic. The integration relies on natural language instructions solely, which is not adequate to represent the complex interaction between the robot and the environment as shown in Li (2025). Points 2 and 3 will be discussed in the next section.

To understand these fundamental limitations, we conducted detailed evaluations on COIN-Primitive and COIN-Composition tasks. The following analysis provides deeper insights into the specific failure modes of each approach.

### 4.3 MAIN RESULTS ON COIN-PRIMITIVE AND COIN-COMPOSITION

**Overview: COIN-Primitive and COIN-Composition reveal specific failure modes.** COIN-Primitive serves as a testbed for evaluating fundamental manipulation skills, while COIN-Composition tests generalization to minor environmental variations. Our detailed analysis confirms the limitations identified in COIN-50 and reveals specific failure modes for each approach category.

**CodeAsPolicy approaches reveal planning-execution disconnects.** Our evaluation confirms the two critical issues identified in COIN-50: (1) *VLM-executor gap*: Significant disconnects exist between high-level VLM planning and low-level execution capabilities. As shown in Figure 4, Voxposer

Table 3: Generalization capability evaluation using COIN-Composition tasks. Models demonstrate severe generalization failures when faced with minor visual or instruction variations from primitive tasks.

| Model | Primitive SR | Composition SR | Finished Tasks | GCS |
|---|---|---|---|---|
| CogACT | 19.0% | 1.5% | 3/20 | 0.079 |
| Pi0 | 16.1% | 6.5% | 4/20 | 0.404 |
| Gr00t N1 | 16.7% | 0.0% | 0/20 | 0.000 |

and Rekep perform poorly even on basic manipulation tasks, indicating fundamental misalignment between planning and execution. (2) Articulation Manipulation problems: These two models are not able to handle the articulated objects, such as cabinet, doors and switchs. This is mainly caused by the structure is not feasible for key-points based representation.

**H-VLA approaches confirm the three limitations identified in COIN-50.** Our detailed analysis reveals specific manifestations of the issues identified earlier: (1) *Poor trajectory and gripper control*: VLAs exhibit severe control precision issues, particularly struggling with gripper timing NVIDIA et al. (2025b). As shown in Table 2, CogACT demonstrates relatively stable trajectories compared to other VLAs, potentially due to its temporal ensemble mechanism, which is left for discussion in future work. However, all VLAs show significant jerky movements and high discontinuity. (2) *Catastrophic generalization failures*: As shown in Table 3, models achieving reasonable success on primitive tasks (16-19%) experience complete failure when faced with composition tasks. Even adding a single new object or switching instructions causes task failure. (3) *Weak VLM-VLA integration*: Despite broader task coverage when overfitted to primitive tasks, the coordination between high-level planning and low-level execution remains fundamentally problematic. For example, while VLAs can successfully execute "open the door" commands, changing the instruction to "pull the door" for the same physical action results in dramatically reduced success rates and moves the gripper to unreasonable locations, demonstrating that the natural language interface fails to capture the underlying action semantics.

## 5 CONCLUSIONS

We present COIN, a systematic evaluation benchmark for interactive reasoning in embodied AI that encompasses three hierarchical evaluation levels: fundamental skill learning (COIN-Primitive), intermediate capability testing (COIN-Composition), and critical interactive reasoning assessment (COIN-50) providing 50 interactive tasks in partially observable settings. Additionally, our COIN-teleoperation pipeline contributes a dataset of 1,000 demonstration trajectories for model training.

Through multi-layered evaluation metrics, our comprehensive analysis reveals fundamental limitations in current EAI approaches, particularly in generalization and adaptability. We provide in-depth analysis of these critical issues across both CodeAsPolicy and H-VLA paradigms. While achieving better interactive reasoning capabilities remains a significant challenge, our findings highlight several promising research directions worthy of deeper investigation:

**Promising Research Directions.** Based on our comprehensive analysis, we identify four critical research directions:

(1) **Improve VLA Trajectory Smoothness**: Our findings suggest that adaptive ensemble mechanisms, as potentially employed in CogACT, may contribute to more stable trajectory control compared to Pi0 and Gr00t.(Table 2). (2) **Enhance Multimodal Perception Ability**: Improving VLA visual generalization and instruction-following capabilities through better multimodal perception could enable more effective VLM-VLA coordination.(Section E.1) (3) **VLM-VLA Integration Mechanisms**: The result showed that H-VLA with natura language is facing great problems according to (Table 3) and Li (2025), showing the necessity to improve the integration between high-level planning and low-level execution, models like Figure AI (2025) might bring more help. (4) **Adaptive CodeAsPolicy Frameworks**: CodeAsPolicy approaches now perform poorly on interactive and primitve tasks, we should adopt adaptive feedback mechanisms to achieve more robust control in dynamic environments (Section 4.2), moving beyond static "plan-then-execute" paradigms.

## 6 ETHICS STATEMENT

Our research presents no significant ethical risks and does not negatively impact human welfare. On the contrary, COIN contributes to the advancement of robotic applications that can assist humans in daily tasks and industrial automation. The benchmark focuses on fundamental manipulation skills in controlled environments, promoting safer and more reliable robotic systems. All experimental data was collected in laboratory settings without involving human subjects or sensitive information.

## 7 REPRODUCIBILITY STATEMENT

To ensure full reproducibility of our results, we commit to open-sourcing all components of our work: (1) Complete source code for COIN benchmark implementation, evaluation metrics, and baseline models will be made publicly available at [https://anonymous.4open.science/r/coin-EB1B/]; (2) The full COIN dataset including 1,000+ demonstration trajectories, task specifications, and environment configurations will be released alongside the code; (3) Detailed experimental protocols, hyperparameter settings, and computational requirements are documented in the appendix to facilitate replication of our findings.

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

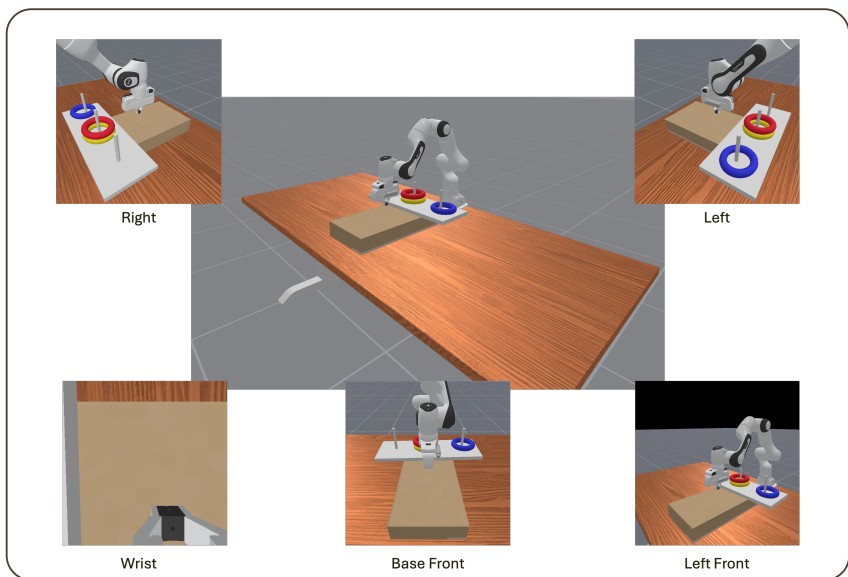

Figure 5: Environment Setup

# A    THE USAGE OF LLM

We acknowledge the use of Large Language Models (LLMs) in the preparation of this work in the following capacities:

**Writing Assistance and Polishing:** LLMs were employed to aid in refining the clarity and coherence of our manuscript. This includes improving sentence structure, enhancing readability, and ensuring consistent academic writing style throughout the paper. All technical content, experimental results, and scientific contributions remain entirely our own work.

# B    LIMITATIONS AND FUTURE WORK

Despite COIN's comprehensive design, several limitations must be acknowledged: (1) our focus on a single robotic platform in static environments fails to capture the full complexity of dynamic real-world scenarios; (2) the absence of dual-arm manipulation tasks that could reveal additional coordination challenges in interactive reasoning.

For future work, we plan to pursue the promising research directions identified in our conclusions. Specifically, we aim to investigate: (1) trajectory smoothness mechanisms inspired by CogACT's temporal ensemble approach to improve VLA execution stability; (2) enhanced multimodal perception architectures that better integrate visual understanding with instruction following; (3) novel VLM-VLA integration paradigms, comparing latent vector bridges against natural language interfaces; and (4) adaptive CodeAsPolicy frameworks incorporating closed-loop feedback for dynamic re-planning. Additionally, we anticipate significant advancement in human-inspired learning approaches that enable iterative "think-execute-think" cycles, allowing models to formulate hypotheses, design informative tests, and recursively update their world models based on interactive outcomes.

# C    ENVIRONMENT SETUP

We use a 7-DoF Franka Emika Panda robotic arm equipped with a parallel gripper as our standard platform. As described in Section 3.1, we use different action spaces for VLA-based controllers and VLM-based planners. Specifically, for VLA models using Panda inverse kinematics, we define the action space as $\Delta p \in [-0.3, 0.3]$ for positional deltas and $\Delta R \in [-0.5, 0.5]$ for orientation deltas. The robot is observed from five camera perspectives providing comprehensive spatial and task

Table 4: Performance comparison on COIN-50 interactive reasoning tasks. Human participants achieve 40% success rate via teleoperation (100% in real-world settings), while current AI approaches rarely exceed 3% success rates, revealing a substantial capability gap that highlights the significant challenges in achieving interactive reasoning.

| Model/Human | Object-centric | Robot-centric | Compositional | Overall |
|---|---|---|---|---|
| **Human (Sim/ 10)** | | N/A | | **40%** |
| **Human (Real/ 10)** | | N/A | | **100%** |
| **H-VLA** | | | | |
| Pi0 + Gemini 2.0 | 1.88% | 2.14% | 1.97% | 1.99% |
| Pi0 + GPT-4o | 1.96% | 2.50% | 2.05% | 2.17% |
| Gr00t N1 + Gemini 2.0 | 1.74% | 2.50% | 1.82% | 2.02% |
| Gr00t N1 + GPT-4o | 1.52% | 1.79% | 1.59% | 1.63% |
| CogACT + Gemini 2.0 | 2.14% | 1.37% | 2.24% | 1.92% |
| CogACT + GPT-4o | 1.74% | 1.07% | 1.82% | 1.54% |
| **CodeAsPolicy** | | | | |
| Voxposer(TD) | 0.43% | 0.00% | 0.45% | 0.29% |
| Voxposer(Normal) | 2.17% | 3.57% | 2.27% | 2.67% |
| Rekep | 3.04% | 3.57% | 3.18% | 3.26% |

context for both VLA and VLM-based planning models. All environments are built on the ManiSkill3 platform using physics-based simulation.

## D    PERFORMANCE COMPARISON: MODELS VS. HUMAN BASELINE ON COIN-50

To contextualize the difficulty level of COIN tasks and establish a performance baseline, we evaluated both current AI systems and human performance on COIN-50 interactive reasoning tasks. For human evaluation, we recruited 3 participants with B.S. degrees who had no prior exposure to our tasks, with each participant attempting a representative subset of 10 tasks twice via teleoperation.

The results reveal a dramatic performance gap between human capabilities and current AI systems. While humans achieve 40% success rates in simulation (100% in real-world settings, confirming task feasibility), the best-performing AI model (Rekep) achieves only 3.26% overall success rate. This 12-fold performance difference demonstrates that current approaches are fundamentally limited in their interactive reasoning capabilities, with substantial gaps that must be addressed before these systems can effectively operate in partially observable environments requiring adaptive manipulation strategies.

## E    VLM COMPARISON

### E.1    VLM PLANNING COMPARISON

We evaluated VLM planners on COIN-50 tasks across execution time steps. GPT-4o consistently outperforms Gemini 2.0 by approximately 1.5 points on our reasoning scale, maintaining this advantage throughout task execution. While they could not improve the performance along the time goes on, indicating there might be some problem on models' historical information utilization.

### E.2    VQA EVALUATION DETAILS

While the end-to-end task execution success rates are low, we evaluate the reasoning capabilities of different VLMs through our embodied VQA protocol. The results reveal that GPT-4o consistently outperforms Gemini-2.0-Flash, and all models show increased accuracy in the middle phases of

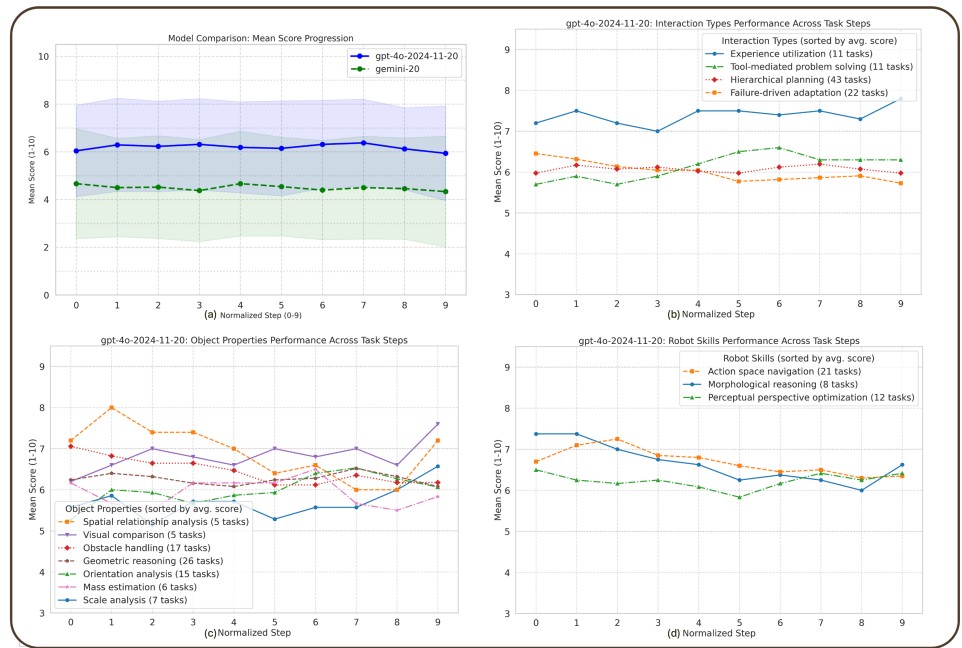

Figure 6: Comparison of VLM reasoning abilities on COIN-50 tasks evaluated along expert demonstration videos. GPT-4o consistently outperforms Gemini 2.0 across different reasoning categories and time steps.

task execution before slightly declining in the final phase. This pattern suggests models gradually accumulate task-relevant information through observation.

## F   MODEL TRAINING CONFIGURATIONS

We use the following training configurations: CogACT-Base trained on $4 \times$ A800 GPUs with device batch size 32 for 30K steps; Gr00t N1 2B trained on $4 \times$ A800 GPUs with device batch size 16 for 120K steps; Pi0-Fast trained on $3 \times$ A800 GPUs with device batch size 2 for 470K steps.

## G   COIN-TELEOPERATION ALGORITHM AND COMPARISON

Our AR teleoperation system demonstrates robust performance across multiple validation metrics:

**Data Quality Validation:** 90% of collected trajectories can be successfully replayed in ManiSkill3, indicating high fidelity data capture. Models trained on our data achieve consistent task performance, validating data quality.

**Cross-Device Compatibility:** The system works reliably across Android and iOS devices released after 2016, with stable 20Hz control frequency maintained even on older hardware (iPhone 7 Plus tested).

**Comparison with Traditional Methods:** Our approach offers significant advantages in accessibility (no specialized hardware required), scalability (easy deployment across environments), and cost-effectiveness (hardware cost under $20 vs. thousands for traditional systems).

## H   MORE DETAILS ABOUT BENCHMARK

### H.1   TASK DIVERSITY AND TEMPORAL ANALYSIS

COIN demonstrates unprecedented temporal complexity compared to existing benchmarks:

---

**Algorithm 1** COIN-teleoperation Pipeline

---

**Require:** Mobile device with sensors and AR framework
**Ensure:** Robot control commands
 1: **function** MOBILEPHONEPROCESSING
 2:     $X_t^{IMU} \leftarrow$ IMU sensor readings at time $t$
 3:     $X_t^{gyro} \leftarrow$ Gyroscope readings at time $t$
 4:     $I_t \leftarrow$ Camera image at time $t$
 5:     $(p_t, R_t) \leftarrow$ AR framework (ARKit/ARCore) processing of $(X_t^{IMU}, X_t^{gyro}, I_t)$
 6:     Establish Web socket connection with PC
 7:     Transmit $(p_t, R_t)$ to PC
 8: **end function**
 9: **function** PCPROCESSINGANDCONTROL
10:     Receive $(p_t, R_t)$ from mobile device
11:     $\Delta p_t = p_t - p_{t-1}, \Delta R_t = R_t \cdot R_{t-1}^{-1}$
12:     $\Delta \hat{p}_t = \text{MedianFilter}(\{\Delta p_{t-9}, \ldots, \Delta p_t\})$
13:     $\Delta \hat{R}_t = \text{MedianFilter}(\{\Delta R_{t-9}, \ldots, \Delta R_t\})$
14:     $\Delta q_t = \text{InverseKinematics}(\Delta \hat{p}_t, \Delta \hat{R}_t)$
15:     Send joint position commands $\Delta q_t$ to robot
16: **end function**

---

| Benchmark | Average Length |
|---|:---:|
| ManiSkill | 52.3 |
| CALVIN | 30 |
| Libero | 77.3 |
| ARNOLD | 125.8 |
| VLABench | 157.2 |
| RLBench | 180.2 |
| RoboCASA Composition | 371.9 |
| **COIN-50** | **988.9** |

Table 5: Trajectory length comparison across benchmarks. COIN features substantially longer temporal horizons, requiring extended reasoning and planning capabilities.

**Multi-Solution Task Diversity:** Over 50% of COIN tasks exhibit substantial procedural diversity with multiple valid solution paths. For example, in Tabletop-Find-Dice, agents can either systematically examine all faces or directly place dice on markers. This diversity prevents simple memorization and requires genuine reasoning capabilities.

**Temporal Dependencies:** At least 40 tasks exhibit strong temporal dependencies where earlier information and actions are essential for later stages, ensuring models must reason over extended horizons beyond local cues.

### H.2 TASK CLASSIFICATION DETAILS

COIN evaluates interactive reasoning across three principal domains—object-centric, robot-centric, and compositional—each capturing distinct yet interdependent aspects of embodied intelligence required for manipulation under partial observability.

### H.3 OBJECT-CENTRIC REASONING

Object-centric reasoning encompasses an agent's capacity to infer and utilize knowledge about environmental entities through strategic interaction:

- **Physical Property Inference (MAS, FRI, SCA, MOV)**: This category examines the agent's ability to:

- *Mass Estimation (MAS)*: Infer object mass distributions and leverage this information to modulate manipulation forces appropriately for optimal handling.
- *Friction Coefficient Assessment (FRI)*: Deduce surface characteristics and their implications for stable grasping and precise manipulation without slippage.
- *Scale Analysis (SCA)*: Assess dimensional compatibility between objects and end-effectors to determine feasible manipulation strategies in space-constrained environments.
- *Moveable Analysis (MOV)*: Determine which objects or object parts can be manipulated and the constraints on their movement, distinguishing between fixed, partially constrained, and freely movable elements.

• **Spatial Reasoning (OBS, GEO, ORI, SRA)**: This domain evaluates the agent's capacity to:

- *Obstacle Handling (OBS)*: Identify which objects constitute impediments in the manipulation space, distinguish between movable and fixed obstacles, and determine whether to relocate obstacles or navigate around them based on task constraints and efficiency.
- *Orientation Analysis (ORI)*: Determine optimal object reorientation for task completion based on geometric and functional constraints in three-dimensional space.
- *Spatial Relationship Analysis (SRA)*: Identify and reason about relative positions between objects, including containment relationships (objects inside other objects), proximity relationships (nearest/farthest objects), and spatial arrangements crucial for task execution.

• **Mechanism Understanding (LOC, KIN, SEQ)**: This category evaluates the agent's ability to:

- *Locking System Comprehension (LOC)*: Deduce the operational principles of locking mechanisms and develop appropriate manipulation sequences to engage or disengage them.
- *Kinematic Constraint Inference (KIN)*: Identify axes of motion in articulated objects to facilitate effective manipulation within their degrees of freedom.
- *Sequential Mechanism Navigation (SEQ)*: Comprehend multi-stage mechanisms and their state-dependent behavior patterns to achieve desired functional outcomes.

• **Visual Reasoning (GEO, VCP, SEM, OCC)**: This category evaluates the agent's ability to process and interpret visual information for manipulation:

- *Geometric Reasoning (GEO)*: Infer shape-based affordances and constraints that influence grasp planning and execution for objects with complex geometries and non-standard forms.
- *Visual Comparison (VCP)*: Compare visual properties across object states or between observed objects and internal representations to detect changes, identify matching features, or recognize anomalies.
- *Semantic Segmentation (SEM)*: Distinguish between different objects or object parts based on visual features, enabling precise targeting of specific components during manipulation tasks.
- *Occlusion Handling (OCC)*: Reason about partially visible or temporarily hidden objects by maintaining object permanence and inferring obscured geometries from limited viewpoints.

## H.4 ROBOT-CENTRIC REASONING

Robot-centric reasoning evaluates an agent's capacity for self-awareness and adaptation within the manipulation context, addressing the embodied nature of interaction:

• **Embodiment Awareness (MOR, PPO, KCA)**: This domain assesses the agent's ability to:

- *Morphological Reasoning (MOR)*: Account for the robot's physical dimensions when planning trajectories and interactions to avoid self-collisions and ensure manipulability.
- *Perceptual Perspective Optimization (PPO)*: Strategically adjust sensor positioning to maximize information gain during task execution, particularly in partially observable environments.
- *Kinematic Constraint Awareness (KCA)*: Reason about joint limitations and workspace boundaries during motion planning to ensure executable action sequences.

• **Control Optimization (DYN, ACT, SKL)**: This category evaluates the agent's capacity to:

- *Dynamic Response Tuning (DYN)*: Adapt control parameters based on task requirements and environmental conditions to achieve desired manipulation outcomes.

    – *Action Space Navigation (ACT)*: Select appropriate actions from the available repertoire based on current state and task objectives within continuous control spaces.

    – *Skill Adaptation (SKL)*: Identify and modify learned manipulation skills to meet specific task requirements and environmental conditions.

### H.5 COMPOSITIONAL REASONING CAPABILITIES

Compositional reasoning encompasses higher-order cognitive functions that integrate multiple reasoning modalities to address complex, interactive challenges requiring adaptive strategies:

- **Tool-Mediated Problem Solving (TOO)**: Identify, create, or adapt tools to overcome manipulation constraints and extend interaction capabilities beyond the robot's native end-effector, enabling solutions to otherwise infeasible tasks.

- **Failure-Driven Adaptation (FDA)**: Actively interact with the environment to gather information about failure modes, then systematically refine strategies based on observed outcomes to develop more robust manipulation approaches through iterative testing.

- **Hierarchical Planning (PLA)**: Decompose complex tasks into coherent sequences of subtasks with appropriate dependencies, adjusting the plan hierarchy in response to changing environmental conditions or task requirements during execution.

- **Experience Utilization (EXP)**: Incorporate historical interaction data into current decision-making processes, applying lessons from previous manipulation attempts to enhance performance in novel but related scenarios through transfer learning.

## I  TASKS TABLE

### I.1  PRIMITIVE TASK

| Task ID | Task Image | Description |
|---------|-----------|-------------|
| Tabletop-Close-Cabinet-v1 |  | Close the cabinet door |
| Tabletop-Close-Door-v1 |  | Close the door |
| Tabletop-Close-Drawer-v1 |  | Close the drawer |

| Task ID | Task Image | Description |
|---|---|---|
| Tabletop-Close-Microwave-v1 |  | Close the microwave |
| Tabletop-Open-Cabinet-v1 |  | Open the cabinet door |
| Tabletop-Open-Door-v1 |  | Open the door |
| Tabletop-Open-Drawer-v1 |  | Open the drawer |
| Tabletop-Open-Microwave-v1 |  | Open the microwave |
| Tabletop-Open-Trigger-v1 |  | Turn on the trigger |
| Tabletop-Pick-Apple-v1 |  | Pick the apple to the marker |

| Task ID | Task Image | Description |
|---|---|---|
| Tabletop-Pick-Book-FromShelf-v1 |  | Find and pick the book from the bookshelf |
| Tabletop-Pick-Bottle-v1 |  | Pick up the bottle and put it on the marker |
| Tabletop-Pick-Cube-ToHolder-v1 |  | Pick up the cube, put it in the holder |
| Tabletop-Pick-Pen-v1 |  | Pick up the pen and put it to the marker |
| Tabletop-Pull-Pivot-v1 |  | Pull the pivot to the target area |
| Tabletop-Put-Ball-IntoContainer-v1 |  | Put the ball into the container |
| Tabletop-Put-Fork-OnPlate-v1 |  | Put the fork on the plate |

| Task ID | Task Image | Description |
|---|---|---|
| Tabletop-Rotate-Cube-v1 | | Rotate the cube till the white face upward |
| Tabletop-Rotate-Holder-v1 | | Rotate the holder till the hole upward |
| Tabletop-Rotate-USB-v1 | | Rotate the USB body for 90 degrees |
| Tabletop-Stack-Cubes-v1 | | Stack all the cubes |

Table 6: Complete COIN-Primitive task specifications with visual examples (20 tasks)

## I.2 INTERACTIVE TASK

| Task ID | Task Image | Description | Obj. | Rob. | Comp. |
|---|---|---|---|---|---|
| Tabletop-Balance-Pivot-WithBalls-v1 | | Put the balls in to the holder to balance the long board on the triangular prism | MAS SCA | no | TOO LPE FDA PLA |
| Tabletop-Clean-For-Dinner-v1 | | Arrange the bowl, fork onto the plate, clean for dinner | no | no | no |

| Task ID | Task Image | Description | Obj. | Rob. | Comp. |
|---|---|---|---|---|---|
| Tabletop-Close-Cabinet-WithObstacle-v1 | | close the cabinet door | OBS | no | PLA |
| Tabletop-Close-Door-WithObstacle-v1 | | close the door | no | no | no |
| Tabletop-Close-Drawer-WithLongObstacle-v1 | | close the drawer | OBS GEO | no | PLA FDA |
| Tabletop-Close-Drawer-WithObstacle-v1 | | close the drawer | OBS GEO | ACT PPO | PLA FDA |
| Tabletop-Find-Book-Black-v1 | | Find and pick the black book from the bookshelf and put it on the marker | GEO OBS | PPO | EXP |
| Tabletop-Find-Book-FromShelf-v1 | | Find and pick the highest book from the bookshelf and put it on the marker | GEO | PPO | EXP |

| Task ID | Task Image | Description | Obj. | Rob. | Comp. |
|---------|-----------|-------------|------|------|-------|
| Tabletop-Find-Cube-RedDown-v1 |  | find the cube which have red face downward, and put it on the marker with red face upward | ORI | PPO ACT | EXP PLA |
| Tabletop-Find-Cube-WithPivot-v1 |  | Move the heavy cube to the goal region | MAS | no | TOO PLA |
| Tabletop-Find-Dice-v1 |  | find the dice which have 2 and 4 point in the corresponding face and put it on the marker | GEO ORI | PPO | EXP PLA |
| Tabletop-Finish-Hanobi-v1 |  | Place all the hanobi in big to small from bottom to up | GEO SEQ | ACT | PLA EXP |
| Tabletop-Insert-Conical-v1 |  | insert the conical to the container | GEO ORI | no | no |
| Tabletop-Insert-Objects-WithShape-v1 |  | insert all the stick on the table into corresponding holes | GEO | no | PLA |

| Task ID | Task Image | Description | Obj. | Rob. | Comp. |
|---------|-----------|-------------|------|------|-------|
| Tabletop-Insert-WithOrientation-v1 | | insert the board on the wall | GEO ORI | no | PLA FDA |
| Tabletop-Keep-Pivot-Balance-v1 | | Balance the long board on the triangular prism and place the cubes to maintain balance | MAS | no | TOO LPE FDA |
| Tabletop-Lift-Book-v1 | | lift the book up to the higher platform | GEO ORI SCA | MOR | PLA |
| Tabletop-Merge-Box-v1 | | Merge ball and boxs up | GEO ORI | no | no |
| Tabletop-Merge-USB-v1 | | Pick up the USB body and insert it into the USB hub | GEO | no | PLA |
| Tabletop-Move-Balls-WithDustpan-v1 | | move all the balls into the holder with dustpan | MAS SCA GEO | no | TOO LPE |
| Tabletop-Move-Balls-WithPivot-v1 | | move all the balls into the dustpan as fast as you can | SCA GEO | no | TOO LPE PLA |

| Task ID | Task Image | Description | Obj. | Rob. | Comp. |
|---------|-----------|-------------|------|------|-------|
| Tabletop-Move-Cross-WithStick-v1 | | Use the stick to move the small cube along the cross-shaped path to the target position | no | no | no |
| Tabletop-Move-Cube-DynamicFriction-v1 | | move the cube to the marker | FRI MAS | no | PLA LPE FDA |
| Tabletop-Move-Cube-WithHolder-v1 | | move the cube to the marker and put the holder on the cube | SCA GEO SEQ | no | PLA |
| Tabletop-Move-Cube-WithPivot-v1 | | move the cube with the pivot to the marker | MAS | ACT DYN | PLA TOO LPE FDA |
| Tabletop-Move-Line-WithStick-v1 | | Use the stick to move the small cube along the straight line path to the target position | GEO ORI | ACT | PLA TOO |
| Tabletop-Open-Cabinet-WithDoor-v1 | | open the cabinet door | OBS | ACT | PLA |

| Task ID | Task Image | Description | Obj. | Rob. | Comp. |
|---------|-----------|-------------|------|------|-------|
| Tabletop-Open-Cabinet-WithObstacle-v1 | | open the cabinet door | OBS | no | PLA |
| Tabletop-Open-Cabinet-WithSwitch-v1 | | open the door, notice the switch will control the state of the door | LOC | no | PLA FDA |
| Tabletop-Open-Door-WithCabinet-v1 | | open the door | OBS | no | PLA |
| Tabletop-Open-Door-WithObstacle-v1 | | open the door | OBS | no | PLA FDA |
| Tabletop-Pick-Cube-Slippery-v1 | | Pick the slippery cube | FRI | ACT | PLA TOO LPE FDA |
| Tabletop-Pick-Cube-WithDoor-v1 | | put the cube to the marker | OBS | KIN ACT | PLA FDA |

| Task ID | Task Image | Description | Obj. | Rob. | Comp. |
|---|---|---|---|---|---|
| Tabletop-Pick-Cube-WithStick-v1 |  | Use the stick to move the small cube along the T-shaped path to the target position | GEO ORI | ACT | PLA TOO |
| Tabletop-Pick-Cylinder-WithObstacle-v1 |  | pick up the center cylinder | LOC | KIN | PLA FDA EXP |
| Tabletop-Pick-Eraser-FromHolder-v1 |  | Pick up the eraser in the holder and place it to the marker | GEO ORI | MOR | PLA EXP |
| Tabletop-Pick-Object-FromCabinet-v1 |  | pick up the object from the cabinet | OBS GEO | PPO MOR ACT | PLA FDA |
| Tabletop-Put-Balls-IntoContainer-v1 |  | move all the balls into the dustpan as fast as you can | GEO | ACT | TOO LPE PLA FDA |
| Tabletop-Put-Cube-IntoCabinetWithObstacle-v1 |  | put the object into the cabinet | OBS GEO | PPO MOR ACT | PLA FDA |

| Task ID | Task Image | Description | Obj. | Rob. | Comp. |
|---------|-----------|-------------|------|------|-------|
| Tabletop-Put-Cube-IntoMicrowave-v1 | | put the cube into the microwave | OBS GEO | ACT MOR | PLA FDA |
| Tabletop-Rotate-Cube-Twice-v1 | | rotate the cube till the green face upward | ORI | ACT | PLA FDA |
| Tabletop-Seek-Holder-InCabinet-v1 | | Find the holder containing an eraser the cabinet, put it to the marker | OBS GEO SEQ | ACT MOR PPO | PLA EXP |
| Tabletop-Seek-Objects-InCabinet-v1 | | Find the apple and the plate in the cabinet, put them on the table | OBS | ACT MOR PPO | FDA PLA EXP |
| Tabletop-Seek-Objects-WithObstacle-v1 | | find the cube in the cabinet and pick it up | OBS | MOR ACT PPO | FDA PLA |
| Tabletop-Slide-Cube-Into-Container-v1 | | Slide a cube down a slope into a container | no | no | no |

| Task ID | Task Image | Description | Obj. | Rob. | Comp. |
|---------|-----------|-------------|------|------|-------|
| Tabletop-Slide-Cube-WithPath-v1 |  | Slide a cube down a slope to the marker | FRI ORI GEO | ACT | PLA |
| Tabletop-Stack-Books-OnBox-v1 |  | Stack all things on the table | ORI GEO | PPO | PLA |
| Tabletop-Stack-Books-v1 |  | Stack all things on the table | SCA ORI | PPO | PLA |
| Tabletop-Stack-Cube-WithColor-v1 |  | Stack the cube with same color | ORI | ACT | PLA FDA |
| Tabletop-Stack-LongObjects-v1 |  | stack all the objects to make it most high | SCA ORI GEO OBS | ACT | FDA PLA |

Table 7: COIN-50 interactive reasoning task specifications with visual examples (50 tasks)

