# OpenReview forum: "COIN: Chain Of INteraction Benchmark: When Reasoning meets Embodied interaction"
_ICLR.cc/2026/Conference — ICLR 2026 Conference Withdrawn Submission_

### Official Review · Reviewer_sYTR · 2025-10-29

**Soundness:** 2
**Presentation:** 1
**Contribution:** 2
**Rating:** 2
**Confidence:** 2

**Summary:**

This work proposes a benchmark designed for interactive reasoning in robotic manipulation. The authors evaluate the current models on the benchmark and show the current models struggle with the interactive reasoning tasks.

**Strengths:**

The paper focus on interactive reasoning which is a practical and important aspect of robotic manipulation task.
The benchmark provides a testbed for evaluating complex interactive reasoning for robotic manipulation.

**Weaknesses:**

The description of the tasks are vague and unclear
	1. for example, why the task is complex so that it requires 1000 step length?  How do we measure the GT planning steps?
	2. There is no clear definition or criteria to differentiate between COIN-50, COIN-primitive and COIN-Composition

Evaluation:
	1. There is no comprehensive evaluation on the interactive reasoning ability, like how well the model can update the reasoning plan given the environment feedback. There is only general discussion about this but lack concrete evaluation which is the key focus of the paper.
	2. The expert demonstration is not used and fully discussed in the paper.

The main contribution of the paper is about the complexity and partial observation of the task, as well as the evaluation.
However, the task diversity, task number and the evaluation scope  make the contribution less significant.

**Questions:**

See weakness above

---

### Official Review · Reviewer_K4JA · 2025-10-31

**Soundness:** 2
**Presentation:** 1
**Contribution:** 2
**Rating:** 2
**Confidence:** 4

**Summary:**

The paper proposes a new benchmark COIN, intended to assess “interactive reasoning” capabilities of SoTA methods that use large models for embodies agent planning and low-level action execution. The authors claim that existing benchmarks do not sufficiently capture this interactive reasoning dimension. To address this, the authors design a suite of 90 tasks (20 primitive tasks, 20 intermediate tasks, and 50 interactive tasks), provide expert demonstrations for each of these tasks, define execution quality metrics, and evaluate end-to-end VLAs, hierarchical VLAs (H-VLA), and CodeAsPolicy methods on these tasks, concluding that existing models struggle with tasks requiring "interactive reasoning."

**Strengths:**

- With large models becoming more capable in embodied domains, it is useful to consider new benchmarks that push beyond static perception or purely language tasks.
- The idea of combining manipulation, obstacles, instructions and perhaps longer horizons is conceptually appealing.
- The authors provide empirical evaluations which show that baseline methods struggle with COIN's "interactive reasoning" tasks.

**Weaknesses:**

- **Lack of Conceptual Clarity on “Interactive Reasoning”**: The central concept of the paper, "interactive reasoning", is never clearly or rigorously defined. The authors claim that prior benchmarks fail to test this capability, but do not concretely illustrate what qualifies as interactive reasoning, how it differs from long-horizon or partially observable manipulation, or which specific tasks uniquely require it. The interactive tasks shown in the appendix do not make this distinction evident. Without a precise definition or motivating examples, the reader cannot meaningfully assess novelty or the relevance of this benchmark.
- **Limited Novelty Relative to Existing Benchmarks**: It remains unclear how the proposed tasks substantially differ from existing embodied AI benchmarks such as Meta-World, CALVIN, RoboCASA, or VLABench. Many tasks described (e.g., “CloseCabinetWithObstacle”) appear conceptually similar to those already available. As written, the novelty of the benchmark over existing work is not convincingly demonstrated.
- **Ambiguities in Evaluation Metrics**: To the best of my knowledge, the metrics presented in the “Fine-grained Execution Quality Metrics” section are non-standard, and are introduced without proper explanation or prior references.
- **Poor Presentation and Figure Quality.**: The paper is difficult to follow. Figures are poorly designed with small text, unclear abbreviations, and sometimes misleading. For instance, Figure 1 shows abstract dialogues instead of the actual simulation environment. Figure 2 is under-explained, making it difficult to understand what each subplot is showing or what each abbreviation means. The text in Figure 3 is very small, and the content is redundant with the explanations in the main text. Tables in the appendix suffer from inconsistent formatting and truncated text. Overall, the presentation quality  severely hinders readability.

**Questions:**

- **Definition of interactive reasoning**: What is the definition of “interactive reasoning” in the embodied VLA context? Could you reference one of COIN's tasks as an examples in the main text to clarify the meaning?
- **Benchmark novelty vs prior work**: Could you provide a more in depth comparison to existing benchmarks and show why your tasks are strictly more challenging (e.g., horizon length, branching decisions, reasoning requirements). If “longer trajectories” is a key claim, please provide normalized statistics (same control frequency, same step definition) and show difference.
- **Metrics clarity**: In the “Fine-grained Execution Quality Metrics” section, could you define each variable/metric you use? Are these metrics common in prior work? If novel, please justify and describe how they correlate with interactive reasoning capabilities (rather than just execution).
- **Baseline performance interpretation**: The results show quite low success even on the “Primitive” task set. Is this expected given the chosen baselines?

---

### Official Review · Reviewer_LEhq · 2025-11-03

**Soundness:** 3
**Presentation:** 2
**Contribution:** 2
**Rating:** 4
**Confidence:** 2

**Summary:**

- The paper introduces the COIN Benchmark to address the gap in evaluating interactive reasoning—the ability to continually interact with a partially observable environment to gather information and update long-horizon plans.

- COIN is structured hierarchically: COIN-Primitive (20 basic skills with 1,000 real-world teleoperated demos), COIN-Composition (20 intermediate tasks), and COIN-50 (50 complex, multi-step tasks). Tasks are systematically categorized into Object-Centric, Robot-Centric, and Compositional Reasoning.

- The paper also develops a low-cost mobile AR teleoperation system for accessible data collection and proposes comprehensive metrics, including a Trajectory Stability Score and a Visual Question Answering (VQA) Score, to evaluate agent performance.

**Strengths:**

- Directly targets the essential and underexplored challenge of interactive reasoning required for practical embodied AI.

 The tiered task structure (Primitive $\rightarrow$ Composition $\rightarrow$ COIN-50) and the principled classification of reasoning abilities facilitate modular skill learning and comprehensive evaluation.

**Weaknesses:**

The author mentions that all VLAs show significant jerky movements and high discontinuity. Is this related to the teleoperation training data provided in this paper? As shown in Table 2, the Human Dataset also achieves poor performance.

This paper is mainly a benchmark article. In the proposed benchmark, the authors demonstrate the current shortcomings of VLA models in terms of motion accuracy and generalization capability. However, this has already been demonstrated by many other works, such as VLABench. Moreover, this paper does not offer any technical contributions, and its evaluation scale is much smaller compared to previous works.

**Questions:**

Please see the weakness section.

---

### Official Review · Reviewer_QzPS · 2025-11-05

**Soundness:** 2
**Presentation:** 1
**Contribution:** 2
**Rating:** 2
**Confidence:** 4

**Summary:**

The paper presents COIN, a benchmark designed to assess interactive causally-dependent reasoning in embodied manipulation. COIN is composed of three parts: COIN-Primitive, covering lower-level base task/skills; COIN-Composition, with mid-level complexity composition of tasks; and COIN-50, for fully complex interactive manipulation tasks in regular-life scenarios.

The paper also describes the data acquisition process using a self-designed low-cost teleoperation system to acquire 1000 demonstrations of its low-level tasks in a simulation setting for a tabletop fixed robot arm with griper. This data is then used across the 3 benchmark level tasks to evaluate different approaches to manipulation: CodeAsPolicy, VLA, and H-VLA.

Presented results in this set of evaluation purport to tease out core model failure modes in the different levels of manipulation tasks.

**Strengths:**

The benchmark addresses some common limitations in recent manipulation benchmarks regarding longer-horizon multi-step manipuation with partial information, often absent from evaluation, or too simplified, causing a mismatch with common real-world try/retrial exploratory needs.

The proposed three levels of manipulation tasks can help focus analysis of results and identify issues at different levels of reasoning and possibly execution.

**Weaknesses:**

Nonetheless, the current submitted paper suffers from some key isuses.

Firstly, the manuscript presentation of tasks and results is very confusing. Results are partially presented without enough context and conclusions stated without details of the results or their analysis. The actual definitions of task themselves only come in Appendix I and even miss the 20 tasks in COIN-composite. Core results of the paper in Table 4 also only appear in Appendix C. The main paper needs at least a basic description of tasks and a proper presentation of results for clarity.

The mentioned results and conclusions, while potentially interesting, don't seem insightful and are nor properly supported by any analysis. Just stated as facts. Without a discussion of the results analysis it is not possible to say if the claims are well derived from the benchmark results.

The paper also claims the low-cost teloperation approach as a contribution. But it doesn't provide any detail on how it actually works, diagrams, or analysis of quality for either usage or data collected.

**Questions:**

The description of the teleoperating system seems to be completely missing. How can cross-device compatibility be claimed if there is only one robot arm settting? If the system can run on different mobile devices is of little relevance to the importance and quality of the tool.

In the discussion of results and conclusions, there are multiple statements potentially interesting. But there is no discussion of analysis or how the conclusions were reached. For example, the paper states "adding a single new object or switching instructions causes task failure", but what's the analysis of its causes? How frequenly does this happen in the experiments? Was there some process to quantify/categorize errors? Or, in discussing VLAs' performance, the paper states they "can successfully
execute open the door commands" but fail if terminology changes. How often do such cases happen? Are there other failure categories? Whatare their frequencies?

Why collect 5 camera views? Which are them and the benefits of the extra trajectory samples? It seems even the experiments only at most use 3.

No files show up for to me when trying the anonymous repo link. I was especially looking for the license under which the benchmark and dataset woulr be released under.

Such benchmarks have a lot of potential, but the current submitted paper needs significant improvements to clarify its findings and contributions.

And a minor issue... COIN is a name clash with the recent "Cheng Chen et al. CoIN: A Benchmark of Continual Instruction tuNing for Multimodal Large Language Model. NeurIPS 2024". Perhaps the authors want to consider a name change.

---

### Note · Authors · 2026-01-16

I have read and agree with the venue's withdrawal policy on behalf of myself and my co-authors.